# The Role of Macrophages in the Pathogenesis of Atherosclerosis

**DOI:** 10.3390/cells12040522

**Published:** 2023-02-05

**Authors:** Alexander V. Blagov, Alexander M. Markin, Anastasia I. Bogatyreva, Taisiya V. Tolstik, Vasily N. Sukhorukov, Alexander N. Orekhov

**Affiliations:** 1Laboratory of Angiopathology, Institute of General Pathology and Pathophysiology, 8 Baltiiskaya Street, 125315 Moscow, Russia; 2Petrovsky National Research Centre of Surgery, 119991 Moscow, Russia; 3Institute for Atherosclerosis Research, 121609 Moscow, Russia

**Keywords:** atherosclerosis, macrophages, polarization

## Abstract

A wide variety of cell populations, including both immune and endothelial cells, participate in the pathogenesis of atherosclerosis. Among these groups, macrophages deserve special attention because different populations of them can have completely different effects on atherogenesis and inflammation in atherosclerosis. In the current review, the significance of different phenotypes of macrophages in the progression or regression of atherosclerosis will be considered, including their ability to become the foam cells and the consequences of this event, as well as their ability to create a pro-inflammatory or anti-inflammatory medium at the site of atherosclerotic lesions as a result of cytokine production. In addition, several therapeutic strategies directed to the modulation of macrophage activity, which can serve as useful ideas for future drug developments, will be considered.

## 1. Introduction

Atherosclerosis is a concomitant disease in coronary heart disease, peripheral arterial pathologies, and cerebrovascular diseases [1]. The pathological basis of atherosclerosis is the accelerated accumulation of plaques that settle in the subendothelial intimal layer of medium and large arteries. This process is caused by the accumulation of cholesterol, predominantly in the form of low-density lipoprotein (LDL) in the intimal layer of the arteries. The result of this accumulation is extended stenosis of the arteries, which sharply reduces blood flow and leads to the development of tissue hypoxia [2]. Severe atherosclerotic complications are stroke and myocardial infarction, the direct cause of which is thrombotic vascular occlusion [3]. The main currently used therapeutic strategy directed to the blockade of the spread of atherosclerotic lesions is the use of drugs that reduce the concentration of LDL cholesterol [2]. At the same time, inflammation is another important pathological process that is directly involved in the development of atherosclerosis. Initially, inflammation is initiated by endothelial cells of blood vessels which react to LDL accumulation. Endothelial cells express elevated levels of cytokines and chemokines that increase inflammation, such as IL 8, CCL2, ICAM-1 and VCAM-1, E- and P- selectin, and others, which are mediators that attract leucocytes to the focus of atherosclerotic lesions [4]. Various cell populations, including dendritic cells, various subtypes of T and B lymphocytes, vascular smooth muscle cells, endothelial cells, and different types of macrophages, are involved in the development of the inflammatory response in the affected intimal layer of the aorta [5]. The latter plays one of the leading roles in the development of the pathological process in atherosclerosis, in addition to inflammation, participating in the assemblage of cholesterol by the formation of foam cells [6]. At the same time, several subtypes of macrophages that participated in the pathogenesis of atherosclerosis are distinguished, and the effect that they provide on inflammation and the further development of atherosclerotic lesions, in general, can be completely different [6]. Therefore, macrophages can be considered in the context of cellular therapeutic targets in atherosclerosis, where the modulation of the functions can result in the weakening of the inflammation and the onset of atheroprotective conditions.

## 2. Significance of Different Phenotypes of Macrophages in the Pathogenesis of Atherosclerosis

### 2.1. M1 Macrophages

Monocytes enter the plaque through processes such as capture, rolling, and transmigration, with each step being regulated by multiple factors. Lipoprotein, especially in its modified forms, accumulates in the proteoglycan-rich intima, where it is taken up by macrophages to form foam cells [7]. Activation of these cells by toll-like receptors (TLRs) as well as interferon-gamma (IFNγ) induced by ingested lipoproteins polarizes them into M1 macrophages. M1 cells secrete high levels of pro-inflammatory factors including interleukin-1β, IL-6, and IL-12; at the same time, they produce, for example, low levels of anti-inflammatory cytokines and chemokines, such as IL-10 and CXCL-9 [8]. As a result of M1 polarization of macrophages, inflammatory factors NF-kB and AP-1 are activated. In this case, both pathways are triggered as a result of exposure to MyD88, which occurs as a response to TLR4 activation. NF-kB is the main regulator of the expression of genes encoding pro-inflammatory cytokines [9]. Macrophages produce various types of Akt proteins that play an important role in macrophage polarization. Thus, it has been shown that Akt2 deficiency is a causative condition that reduces the ability of macrophages to polarize toward the M1 phenotype [10]. Using a mouse model of ApoE^−/−^, it was demonstrated that deletion of Akt1 is associated with the progression of atherosclerotic lesions and the predominant polarization of macrophages by the M1 type [11]. It is known that activated M1 macrophages are the dominant producers of reactive oxygen species (ROS) through the enzyme NADPH oxidase, which triggers the reaction of chronic inflammation and the formation of atherosclerotic plaques [12]. In addition, it has been shown that M1 macrophages are the most common cell population in the lesion in patients with coronary heart disease [7].

### 2.2. M2 Macrophages

In the same way, M2-activated macrophages act at the sites of atherosclerotic lesions, contributing to the reduction of inflammation, removal of cellular debris after apoptosis, and tissue repair [13]. M2 macrophages form three groups—M2a, M2b, and M2c macrophages—which are triggered in response to various stimuli. Thus, the initiating beginning for macrophages of the M2a subtype is an increased concentration of anti-inflammatory cytokines IL-4 and IL-13 [14]. They produce profibrotic factors such as fibronectin and TGF-β, which promote tissue regeneration [1]. Macrophages of the M2b subtype are triggered by TLR ligands or IL-1 receptor agonists. This macrophage subtype secretes both anti-inflammatory and pro-inflammatory cytokines. Macrophages of the M2c subtype are capable of being activated in response to the production of IL-10 and glucocorticoids [15]. All M2 macrophage subtypes show an anti-inflammatory cytokine profile characterized by low production of IL-12 and high production of IL-10. At the same time, it remains unclear what specific roles each of the M2 macrophage subtypes play in pathogenesis [16].

In in vivo studies of mice models of atherosclerosis, it was shown that with the regression of the disease, the content of M2 markers in macrophages in atherosclerotic plaques increased, while M1 markers, on the contrary, decreased. Atherosclerotic lesions containing high levels of M2 macrophages have a reduced inflammatory response, have faster recovery, and contain low plaque cholesterol concentrations [17]. The introduction of the cytokine IL-13, which induces the transition of macrophages from the M1 to the M2 phenotype, promotes the regression of atherosclerotic lesions as a result of an increase in the concentration of collagen in the lesions and a decrease in VCAM-1 mediated recruitment of monocytes. In addition, M2 macrophages are capable of phagocytosis of macrophages of the M1 phenotype, which also leads to a decrease in inflammation and inhibition of the formation of a necrotic nucleus in the foci of atherosclerotic lesions [18].

### 2.3. Other Macrophage Phenotypes

In addition to the M1 and M2 phenotypes indicated above, there are a number of other macrophage phenotypes that can also play a different role in the development of inflammation and the progression of atherosclerosis. The general scheme of macrophage types involved in the pathogenesis of atherosclerosis and their activation factors is shown in Figure 1.

M1 macrophages are the main inflammatory type of macrophages. MMe (Metabolic activated) macrophages are predominantly located in adipose tissue and their main function is the removal of dead adipocytes. Mox macrophages are inflammatory macrophages which produce a high level of the Hmox1 (Heme Oxygenase 1) enzyme. M1, MMe, and Mox macrophages are activated by LDL (Low density lipoproteins) and interferon gamma (IFNγ). M4 macrophages are the inflammatory type of macrophage; they maturate and are activated by platelet chemokine CXCL-4. M1 macrophages are the main anti-inflammatory type of macrophages; they are activated by anti-inflammatory chemokines IL4 and IL13. HA-mac, M ((Hb)–hemoglobin stimilated), and Mhem macrophages are anti-inflammatory macrophages that have expressed an atheroprotective effect; they are activated by hemoglobin-haptoglobin protein (hb-hp) complex, which is involved in the clearance of hemoglobin in hemorrhagic areas.

Thus, the phenotypes of MMe and Mox macrophages are atherogenic; their maturation in adipose tissue occurs under the action of pro-inflammatory cytokines and oxidized phospholipids, which are formed from oxidized low-density lipoproteins (ox-LDL), which play an important role in inflammation chronification [19]. MMe macrophages are characterized by high activity of the enzyme NADPH oxidase-2 and iNOS, which play a significant role in inflammation, and the generating of reactive oxygen species (ROS). In turn, the Mox phenotype is associated with the expression of surface markers such as Srnx-1 and Txnrd-1; this type of macrophage is widely found in developed atherosclerotic plaques [20].

HA-mac and M (Hb) macrophages are atheroprotective phenotypes [21]. They are activated by the hemoglobin-haptoglobin protein (hb-hp) complex, which is involved in the clearance of hemoglobin in hemorrhagic areas. One of the functions of M(Hb) macrophages in the pathogenesis of atherosclerosis is associated with the induction of cholesterol efflux, which results in a sharp decrease in foam cells [21]. Another atheroprotective macrophage phenotype whose activation is also associated with the hemoglobin-haptoglobin complex is Mhem. In addition to the involvement of Mhem in erythrophagocytosis, this macrophage phenotype inhibits the development of oxidative stress, the accumulation of lipid droplets, and, accordingly, the formation of foam cells [22].

M4 macrophages are activated by CXCL-4, which occurs in human atherosclerotic plaques. Macrophages M4 have proinflammatory and proatherogenic properties, which are strongly manifested in arterial thrombosis. They can participate in the degradation of the fibrous cap and plaque rupture due to the production of the MMP12 enzyme [22].

### 2.4. Molecular Mechanisms of Macrophage Phenotypic Shift

IRF/STAT signaling is the main signaling pathway that determines the direction of macrophage polarization. IFNγ binds to its receptor on the surface of the macrophage membrane and activates the STAT1 factor; LDL binds to TLR4, which leads to the activation of the factors NF-κB and IRF3. In both cases, the launch of these cascades contributes to the shift of the macrophage phenotype towards M1 [9]. In turn, anti-inflammatory cytokines such as IL-4 and IL-13, binding to their receptors, activate a number of factors, such as IRF4, STAT6 and PPARγ, and IL-10-STAT3, which contributes to the polarization of macrophages towards M2 [9]. In addition, the polarization of macrophages towards the pro-inflammatory type can be affected by the state of hypoxia that occurs in atherosclerotic plaques. It is known that the hypoxia factor HIF-1α is able to activate the inflammatory factor NF-κB and increase the expression of NO synthase, which leads to a shift in the macrophage phenotype towards M1 [9]. In addition, a number of miRNAs have been identified that contribute to the polarization of macrophages. Thus, miR-155 is capable of shifting the phenotype towards M1 by suppressing the expression of IL-13Rα1 [23]. Accordingly, miR-let-7i and miR-let-7b reverse bias towards the M2 phenotype by downregulating TLR4 expression [24].

### 2.5. Distribution of Macrophages in Atherosclerosis

It has been demonstrated that different macrophage phenotypes localize in different sites of atherosclerotic plaque. It is known that M1 macrophages are the dominant phenotype in progressive plaques and that M2 macrophages predominate in stable plaques in which inflammation resolves and in vascular adventitia [25]. In the intima of vessels, M1 macrophages transform into foam cells which support the development of inflammation and further polarization of macrophages towards M1. At the same time, there is an equal proportion of M1 and M2 macrophages in fibrous capsules. With increased atherosclerotic lesions, the formation of a necrotic nucleus occurs, which results in a subsequent increase in inflammation and necrosis. M1 macrophages are the dominant population in the lipid core near the necrotic core, while M2 macrophages are peripherally concentrated in areas of neovascularization but can approach M1 macrophages and phagocytose them, resolving inflammation [26]. Atherogenic M4 macrophages are predominantly localized in the adventitia and intima; Mox macrophages are located near the lipid core [26]. Atheroprotective macrophages M (Hb) and Mhem are localized predominantly in areas of neovascularization in atherosclerotic plaques [27].

## 3. Lipid Activation of Macrophages in Atherosclerosis

It is known that LDL is the main source of lipid deposits in the arterial intima in atherosclerosis, whereas the cause of the increase in the concentration of intracellular cholesterol is modified LDL [28,29]. It is known that various forms of LDL are able to induce polarization of macrophages towards the M1 phenotype. In a study on macrophage cell culture, it was found that the introduction of LDL led to an increase in the expression of pro-inflammatory cytokines TNF-α and IL-6 and a decrease in the expression of CD206 and CD200 receptors, which are secreted mainly by M2 macrophages [30]. The polarization of macrophages is strongly influenced by oxidized forms of LDL, which are recognized through toll-like receptors (TLR) and scavenger receptors, which triggers a cascade of inflammatory response and transformation of macrophages towards an inflammatory phenotype [31]. It was noted that TLR activation is also associated with an increase in the activity of protein kinase C and NADP oxidase 2 enzymes. As a result, an increase in the expression of pro-inflammatory cytokines occurs in macrophages. In the sum of changes caused by the action of oxidized LDL on macrophages primarily associated with an increase in the expression of pro-inflammatory cytokines and a decrease in the expression of anti-inflammatory cytokines, macrophages are polarized towards an atherogenic phenotype [32,33].

It is known that phospholipase-induced hydrolysis of lipoproteins, the consequence of which is the release of free phospholipids and fatty acids, can be carried out in an environment with an acidic pH. This is directly related to the accumulation of lipids in the arterial wall and the development of atherosclerotic plaques. The effect of LDL on macrophages results in the production of pro-inflammatory cytokines TNF-α and IL-6. The pro-inflammatory environment promotes the formation of foam cells [34]. Pro-inflammatory phospholipid and fatty acid signaling is mediated by the G-protein-coupled G2A receptor, which plays an important role in the pathogenesis of the disease since its deficiency leads to severe atherosclerosis and the acquisition of a pro-inflammatory M1 phenotype by macrophages. Saturated fatty acids also promote pro-inflammatory macrophage polarization through the activation of the TLR-NF-κB signaling cascade [35].

It is reliably known that polyunsaturated fatty acids (PUFAs) have pronounced atheroprotective properties, which, among other things, are associated with their anti-inflammatory effect on macrophages. Thus, linoleic acid can suppress the expression of pro-inflammatory genes in macrophages, including NF-kB, CCL2, and cyclooxygenase 2 through PPARγ receptors, which reduces the progression of atherosclerosis. In addition, PUFAs are able to show a counteracting effect on the atherogenic effects of saturated fatty acids, such as the palmitate-induced expression of the lectin-like oxidized LDL-1 receptor (LOX1) [34].

The atheroprotective functions of high-density lipoproteins (HDL) are associated with the stimulation of catabolism and the outflow of cholesterol. A notable phenomenon observed in patients with atherosclerosis is a decrease in the relative level of HDL compared to LDL [36]. At the same time, the atheroprotective effect of HDL is partially associated with an anti-inflammatory effect; in studies on a mouse model, it was shown that HDL contributes to macrophage polarization from M1 to M2 phenotype and inhibits backward polarization towards M1 [37]. Lipid modulation of pro- and anti-inflammatory macrophage phenotypes may be a promising therapeutic strategy in the treatment of atherosclerosis [38].

## 4. Foam Cell Formation and Their Role in Disease Progression

Intracellular lipid accumulation is one of the early events in the development of atherosclerosis. The formation of foam cells from macrophages is associated with the suppression of LDL receptor expression, which allows these cells to internalize apoB- containing lipoproteins. Modified LDL, which is internalized by alternative mechanisms, is the main source of cholesterol accumulation in foam cells, as shown in studies in vitro [34]. Oxidation is the most studied atherogenic modification of LDL. There is a hypothesis that high oxidative stress is the cause of the formation of atherogenic oxidized LDL, which alone can induce the development of inflammation in the arterial wall and dyslipidemia. When studying the composition of LDL in patients with atherosclerosis, various molecular modifications were revealed, including glycation, desialylation, and complex formation [39]. It is noted that complex formation is the most atherogenic modification of LDL. As a result of passage through the subendothelial layer of the arterial wall, modified LDL binds to proteoglycans; thus, keeping them in the arterial wall and leading to lipid accumulation. Most likely, this is where the LDL modification process ends [35].

Scavenger receptors play an important role in the recognition of modified LDL by macrophages, among which SRA1, CD36, and LOX1 are of the greatest importance in the development of atherosclerosis. Deficiency of these receptors may reduce foam cell formation in Apoe^−/−^ mice, thereby suggesting that macrophages have other LDL uptake mechanisms. A significant accumulation of native forms of LDL, which are formed during hyperlipidemia in growing plaques, can also induce the appearance of foam cells [40,41].

A result of the ultrastructural analysis of macrophages incubated with modified LDL was the demonstration of the accumulation of LDL in lysosomes. Biochemical studies have shown that after internalization, LDL particles are cleaved in lysosomal compartments to free cholesterol and fatty acids, and free cholesterol is transported to the endoplasmic reticulum (ER), where it undergoes re-esterification [42]. Accumulation of cholesterol in ER membranes leads to its defective esterification by ACAT1 and a further increase in accumulation. In addition, ER stress associated with cholesterol accumulation in macrophages also takes a role in the disease development by increasing apoptosis in developing atherosclerotic plaques, in which increased apoptosis and impaired clearance of dying cells lead to the formation of a necrotic core. Membrane microdomains, rich in cholesterol, facilitate pro-inflammatory signaling mediated by TLR and NF-κB [43].

Recent studies have also demonstrated the involvement of vascular smooth muscle cells (VSMC) in the formation of foam cells. Smooth muscle-mesenchymal cells (SMMCs) undergo transdifferentiation and become macrophage-like cells during atherogenesis [44]. Foam cells, which are derived from SMMC, are phenotypically transformed with the loss of contractile markers in atherosclerotic plaques [45]. This phenotype-switching phenomenon of SMMC has also been confirmed in humans. Approximately 40% of macrophages in plaques originate from SMMCs, which are the product of the phenotypic transformation of SMMCs [46]. Therefore, inhibition of the phenotypic transformation of HCMC is useful for suppressing the increasing amount of macrophages in the plaque and maintaining the amount of SMMC in the fibrous cap. It has been confirmed that as a result of exposure to inflammatory mediators, SMMCs undergo transformation into macrophage-like cells. This is confirmed by the high expression of macrophage markers and the low content of smooth muscle cell markers. [47,48].

The formed foam cells cause disruption of efferocytosis and initiate various pathways of cell death such as apoptosis, pyroptosis, and necroptosis. An increase in the number of dead cells leads to the formation of a large number of necrotic nuclei, which destabilizes atherosclerotic plaques.

## 5. Importance of Cytokines Produced by Macrophages in Atherosclerosis

### 5.1. Pro-Inflammatory Cytokines

Macrophages release cytokines such as IL-1, IL-6, IL-8, IL-12, IL-18, soluble CD40L (ligand of CD40 receptor), and TNF in response to various inflammatory stimuli. The IL-1 family includes 11 proteins such as IL-1α, IL-1β, IL-1 receptor antagonist (IL-1RA), IL-18, and other less studied cytokines [49]. IL-1α and IL-1β are pro-inflammatory cytokines produced by myeloid cells. The secretion of cytokines of the IL-1 family and the expression of their receptors are increased in atherosclerotic aortas [50]. IL-1β is an important Th17 cell differentiation factor that can enhance inflammation in the vascular wall. In vivo studies have confirmed the role of IL-1α and IL-1β as atherogenic agents that increase the expression of adhesion molecules and activate macrophages [51].

It has been proven that the expression of the pro-inflammatory cytokine IL-18 increases in atherosclerotic plaques as well as in samples taken from patients with diabetes mellitus and myocardial infarction [52]. In a mouse model study in Apoe^−/−^ mice, administration of IL-18 led to the progression of atherosclerosis in experimental animals, and the effect of its endogenous inhibitor was associated with an improvement in the condition of animals and resolution of inflammation [53].

The activity of the cytokine IL-6 is noted in the pathogenesis of many inflammatory diseases. Its receptor is both soluble IL-6R and membrane-bound [54]. In studies with the introduction of the monoclonal antibody canakinumab directed against IL-1β, an impairment of IL-6 signaling was also observed, which was associated with a reduced risk of developing cardiovascular complications and occurred regardless of the development of hypolipidemia [55]. It is also noted that IL-6 is able to accelerate the development of atherogenesis as a result of changes occurring due to the aging of the body. It follows that inhibition of IL-6 activity is a promising strategy in the treatment of atherosclerotic lesions, especially in elderly people [55].

It is known that IL-8 is a pro-inflammatory cytokine that causes an atherogenic effect in atherosclerosis; it is mainly secreted by macrophages. As a result of IL-8 activation in the affected vessels, macrophages are recruited with their subsequent coagulation in the aortic intima [56].

Production of IL-12 is carried out by macrophages in atherosclerotic plaques. This pro-inflammatory cytokine induces the maturation of Th1 T helper cells through the activation of IL-12Rβ2. Th1 CD4+ T-lymphocytes are able to additionally stimulate macrophages and, accordingly, the further development of inflammatory reactions. In a study in ApoE^−/−^ mice, it was demonstrated that IL-12 administration was the cause of the growth of atherosclerotic lesions in animals [54].

CD40 production is mainly carried out by M1 macrophages. Plasma CD40 concentration has been shown to be closely correlated with the severity of the state of atherosclerotic lesions in the carotid arteries as well as the existing risk of potential cardiovascular complications. Plaque CD40 and CD40L levels are directly related to vascular remodeling. Thus, both CD40 and CD40L are directly involved in the progression of atherosclerosis [57].

The functions of TNF-α in atherosclerosis are associated with the activation of endothelial cells, which leads to an increase in the expression of various adhesion proteins (like E- and P-selectins, VCAM-1, ICAM-1, and PECAM-1) [58,59,60], activation of the production of other pro-inflammatory cytokines, and the recruitment of immune cells to the affected arterial intima. It is known that TNF-α works through interaction through two receptors: TNFR1 and TNFR2. Increased expression of TNF-α, which is activated as a result of signaling through the regulatory proteins LOX-1 and NF-kB, in turn causes hyperproduction of ROS, which is one of the causes of endothelial dysfunction in a number of cardiovascular diseases [61].

### 5.2. Anti-Inflammatory Cytokines

One of the main anti-inflammatory cytokines is IL-10, whose properties also determine the atheroprotective effect. In a mouse model, an inverse relationship was shown between the severity of atherosclerosis and IL-10 production by B cells, and it was also found that high cholesterol levels can mask the revealed atheroprotective properties of IL-10 + B-lymphocytes [62]. In another study, PDCD4-deficient mice with an induced increase in IL-10 expression attenuated the progression of atherosclerosis [63].

It is known that the effect of TGF-β on atherosclerosis can have both atherogenic and atheroprotective effects [64]. Thus, a deficiency of the GDF15 protein, which is a member of the TGF-β protein family, leads to a decrease in the growth of atherosclerotic formations and slowing macrophage chemotaxis in Ldlr mice. It has also been shown that TGF-β is able to suppress pro-inflammatory signals from endothelial cells in the arterial wall, which, in addition to reducing inflammation and vascular permeability, led to the regression of the disease in mice. However, as indicated, TGF-β is also able to exhibit atherogenic properties, which is associated with its effect on smooth muscle cells even at an early stage of the development of atherosclerotic plaques [29].

## 6. Therapeutic Strategies Targeting Macrophages in Atherosclerosis

### 6.1. Induced Macrophage Polarization

As discussed in the previous section, all macrophage subtypes can be divided into two groups depending on the pathological effect in the development of atherosclerosis: the first group has a pro-inflammatory effect and an atherogenic effect, and the second group has an anti-inflammatory effect and an atheroprotective effect. However, at the same time, as a result of exposure to a number of microenvironmental stimuli, some macrophage subtypes can transform into other subtypes. This plasticity can be induced, thus increasing the proportion of anti-inflammatory macrophages and decreasing the proportion of pro-inflammatory ones. For a number of animal models, it was shown that the introduction of anti-inflammatory factors, such as HDL; anti-inflammatory cytokines: IL-4, IL-13, and IL19, promote the transformation of macrophages from the M1 subtype to the M2 subtype, while reducing the progression of atherosclerotic lesions [27]. A number of natural medicinal compounds, such as curcumin, ginsenosides Rb1 and Rg3, polyphenols, and melatonin can induce the polarization of macrophages [27]. In addition, this ability has been shown for synthetic drugs such as sitagliptin and metmorphine [27]. The discovery of the ability to induce polarization of macrophages towards an anti-inflammatory phenotype for compounds registered as drugs may contribute to a faster implementation of this strategy in clinical practice if this technique is successfully tested in preclinical experiments.

### 6.2. Inhibition of Scavenger Receptor Activity

One of the main triggers in the inflammatory response in atherosclerosis is the recognition of modified LDL by macrophages through scavenger receptors (SRs). Accordingly, induced inhibition of the binding of modified LDL to macrophage SR can block inflammation even at the early stages of its development. Thus, in the study [65], glycosylated micelles were used, which competitively blocked the binding of SRs MSR1 and CD36 to LDL, which reduced the accumulation of LDL. In addition, for a number of amphiphilic macromolecules that are able to selectively bind to SR, therapeutic efficacy in reducing cholesterol levels and general inflammation in atherosclerosis was shown in some animal models [66]. Another study showed that the introduction of synthetic proteins with a high affinity for CD36 reduced the uptake of oxidized LDL by macrophages and the number of atherosclerotic plaques in an APOE-negative mouse model [67].

### 6.3. Autophagy Induction in Macrophages

Autophagy is a catabolic cellular process during which dysfunctional proteins and organelles are degraded, thus clearing the cell of unnecessary components. It has been shown that the formation of atherosclerotic plaques increases in mice deficient in the ATG5 gene, which encodes the main regulator of autophagy [68]. Since there is a decrease in the production of reactive oxygen species during mitophagy, as well as inhibition of the activation of the NLRP3 inflammasome and a number of other inflammatory pathways [69], and macrophages are the main initiators of inflammation, a decrease in inflammatory stimuli in macrophages can potentially inhibit the development of inflammation in atherosclerosis. A number of medicinal compounds capable of inducing macrophage autophagy, such as simvastatin, berberine, and ursolic acid, have been shown to have a positive effect in reducing atherosclerotic lesions [68]. The general outline of the described therapeutic strategies is shown in Figure 2.

## 7. Discussion

In the context of the introduction of therapeutic strategies aimed at modulating macrophage function in atherosclerosis, the use of these strategies for the treatment of other diseases whose pathogenesis is directly related to the activity of macrophages is also considered and potentially promising, for example, for diseases such as chronic kidney disease [70], rheumatoid arthritis [71], and a number of other chronic inflammatory diseases. In addition, the level of expression of cytokines and chemokines produced by various types of macrophages may possibly indicate the level of progression or the slowing down of atherosclerosis. The study of the correlation between the level of molecular mediators released by macrophages and the clinical picture of atherosclerosis may be the first step towards the creation of convenient diagnostic tools that more accurately assess the condition of patients. Interestingly, lipoproteins are able to directly induce macrophage activity, as noted at the beginning of the review. In this regard, it is important to study how M1-macrophage activation depends on the type of LDL modification. It can identify the most “aggressive” LDL types in atherosclerosis. An important direction in the fight against atherosclerosis is also the search for new markers of the development of atherosclerosis that are associated with the activity of macrophages. Thus, the study [72] revealed a new marker secreted by macrophages—the YKL-40 protein—whose serum level directly correlated with the risk of developing major cardiovascular outcomes (MACE) in patients with arterial hypertension. The study [73] identified several genes that express markers of atherosclerosis progression in M1 macrophages. Thus, it was found that ALOX5 and NCF2 are able to participate in the formation of the necrotic nucleus through the induction of ferroptosis of macrophages. In order to identify new markers of atherosclerosis, an analysis of the gene expression profile was carried out in the study [74], where several markers were found associated with the migration of immune cells, including macrophages, into the lesion in atherosclerosis. Since, in addition to macrophages, many other types of immune cells are involved in the pathogenesis of atherosclerosis, the study of their influence on the chronification of inflammation and atherogenicity is also important in the fight against this disease. This is especially true of immune cells, for which it is unclear what role they play in the progression of atherosclerosis, such as some subpopulations of T-lymphocytes [75].

## 8. Conclusions

Macrophages are one of the key groups of immune cells involved in the pathogenesis of atherosclerosis. Moreover, all macrophage phenotypes can be divided according to their effect on the pathogenesis of atherosclerosis into pro-inflammatory atherogenic, which contribute to the progression of atherosclerosis, and anti-inflammatory atheroprotective, which weaken the course of the disease. The action of macrophages is directly induced by lipoproteins: HDL contribute to the activation of atheroprotective forms of macrophages, and LDL cause the activation of atherogenic macrophages. Modulation of macrophage activity can be considered as one of the promising therapeutic pathways in the treatment of atherosclerosis, which can be divided into several strategies: induced macrophage polarization, inhibition of scavenger receptor activity, and induction of autophagy in macrophages.

## Figures and Tables

**Figure 1 cells-12-00522-f001:**
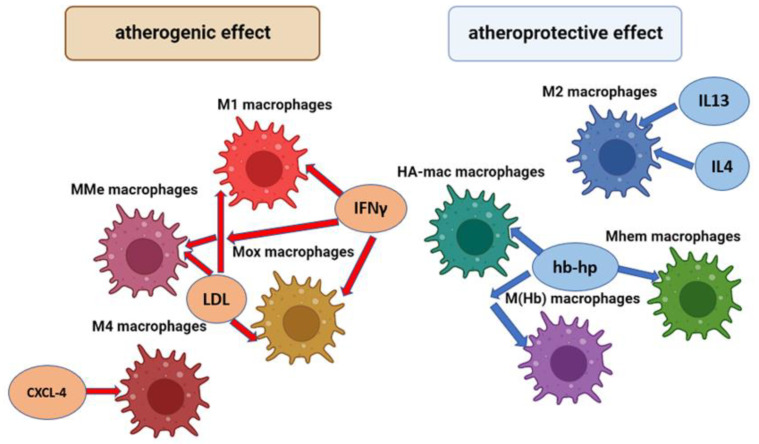
Types of macrophages involved in the pathogenesis of atherosclerosis and their factors of activation.

**Figure 2 cells-12-00522-f002:**
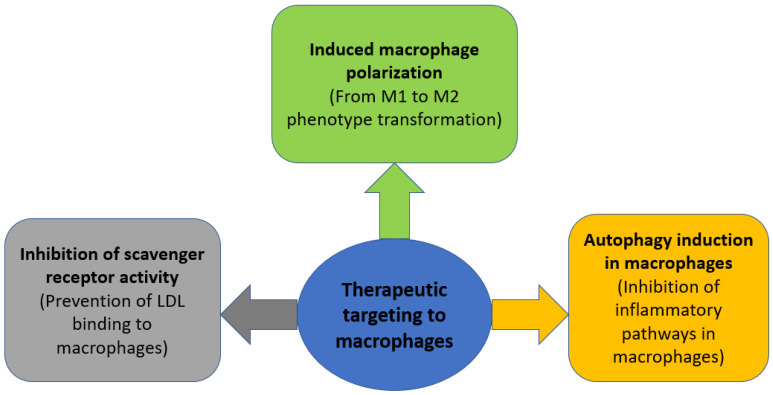
Therapeutic strategies aimed at modulating macrophage activity in atherosclerosis.

## Data Availability

Not applicable.

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
