# Peer review of "The Role of Macrophages in the Pathogenesis of Atherosclerosis"

_cells, 2023, doi:10.3390/cells12040522_

Round 1

Reviewer 1 Report

This review on the roles of macrophages in the pathogenesis of atherosclerosis contains useful, important information for investigators in the field. However, it can be improved by broader coverage, clearer statements and better word choice. Additionally, throughout the review, there are some reports of study results without references. 

Figure 1, as presented, provides very little information about the types and properties of macrophage. The legend should explain the abbreviated names – Mme, Mox, etc  -  and provide their properties.

Specific comments:

Line 35 – ‘taking?’ (use).     Lines 52/53 – ‘..functions (which can result in weakening of) inflammation and the (onset) of …’.      Line 70 –‘..causative factor? (condition)..’.     Line 82 – delete ‘including’.     Line 108 – Suggest placing (Figure 1) right after ‘atherosclerosis.’     Lines 112/113 – NADPH oxidase-2 and iNOS are both generators of reactive oxygen species (oxidants) in M1-like macrophages.    Line 117 – suggest inserting [reference] after ‘phenotype’.     Line 129 – suggest deleting the last sentence of paragraph.    Line 141 – remove ‘R’ from ‘CD200R’?    Line 143 – define ‘TLR’.     Lines 154-157 – unclear sentence.   Lines 174-177 – confusing sentence.   Lines 213/215 – define ‘SMMC and SHMC’.   Line 219/220 – define ‘CBM’.     Line 231 – define CD40L (ligand).  Lines 281-284 - what is the reference for this study?  Line 286 – a reference is needed after the first sentence.  Line 287-289 – confusing sentence; ‘which leads to a (pronounced decrease in) macrophage..’    Line 310 – melatonin is a naturally occurring compound.   Line 316 – omit ‘triggering’.  Line 319-321 – confusing sentence.    Lines 355-360 – long and confusing; shorter, more direct sentences needed. 

Author Response

Point 1: Figure 1, as presented, provides very little information about the types and properties of macrophage. The legend should explain the abbreviated names – Mme, Mox, etc  -  and provide their properties.

Response 1: Figure 1 was improved. The legend was added.

Point 2: Line 35 – ‘taking?’ (use).     Lines 52/53 – ‘..functions (which can result in weakening of) inflammation and the (onset) of …’.      Line 70 –‘..causative factor? (condition)..’.     Line 82 – delete ‘including’.     Line 108 – Suggest placing (Figure 1) right after ‘atherosclerosis.’     Lines 112/113 – NADPH oxidase-2 and iNOS are both generators of reactive oxygen species (oxidants) in M1-like macrophages.    Line 117 – suggest inserting [reference] after ‘phenotype’.     Line 129 – suggest deleting the last sentence of paragraph.    Line 141 – remove ‘R’ from ‘CD200R’?    Line 143 – define ‘TLR’.     Lines 154-157 – unclear sentence.   Lines 174-177 – confusing sentence.   Lines 213/215 – define ‘SMMC and SHMC’.   Line 219/220 – define ‘CBM’.     Line 231 – define CD40L (ligand).  Lines 281-284 - what is the reference for this study?  Line 286 – a reference is needed after the first sentence.  Line 287-289 – confusing sentence; ‘which leads to a (pronounced decrease in) macrophage..’    Line 310 – melatonin is a naturally occurring compound.   Line 316 – omit ‘triggering’.  Line 319-321 – confusing sentence.    Lines 355-360 – long and confusing; shorter, more direct sentences needed.

Response 2: All was corrected.

Reviewer 2 Report

Macrophages are highly heterogeneous and plastic cells. Different subtypes of macrophages can either promote or suppress a variety of responses, including inflammation, oxidative stress, cell clearance, wound healing, and autoimmunity. This article reviews the role of macrophages in the pathogenesis of atherosclerosis, I do not feel that it addresses a gap in the literature. The only difference in this review is the significance of different phenotypes of macrophages in the progression or regression of atherosclerosis. However, the molecular mechanisms underlying macrophage phenotypic shift were not summarized, and the distribution of macrophage subsets within different stages of atherosclerosis and how these are mechanistically related to cardiovascular (CV) vulnerability were not discussed. In addition, the current knowledge/evidence on macrophage subtype-specific markers as predictors of risk of CV events should be discussed and updated in the review.

Author Response

Point 1: However, the molecular mechanisms underlying macrophage phenotypic shift were not summarized, and the distribution of macrophage subsets within different stages of atherosclerosis and how these are mechanistically related to cardiovascular (CV) vulnerability were not discussed.

Response 1: The sections: 2.4 “Molecular mechanisms of macrophage phenotypic shift” and 2.5. “ Distribution of macrophages in atherosclerosis” were added.

Point 2: In addition, the current knowledge/evidence on macrophage subtype-specific markers as predictors of risk of CV events should be discussed and updated in the review.

Response 2: The information about new studies in this area was added in Discussion (lines: 500-511).

Reviewer 3 Report

The review is well organized and the topics are presented in a very clear way. In my opinion, the English language and style are adequate. Overall, my opinion is positive. I only have some minor suggestions for the authors, aimed at further improving this review, which is already a good quality paper.

1) Throughout the manuscript, sometimes "in vivo" and "in vitro" are in italic font, some other times don't. I suggest the authors to align the style.

2) To enhance the summarising function of figure 1, I suggest the authors to add some more information next to each type of macrophages, in a schematic way. For example, which cytokines are most secreted, what triggers the activation of that particular type of macrophage, and so on. 

3) I suggest the authors to increase the font of figure 2, in order to ease the reading.

4) On lines 272-274, when the authors state: "...which leads to an increase in the expression of various adhesion proteins, activation of the production of other pro-inflammatory cytokines, and the recruitment of immune cells to the affected arterial intima.", I suggest to mention some examples of adhesion molecules involved in these processes. Therefore, I suggest the authors to replace the sentence with: "...which leads to an increase in the expression of various adhesion proteins (like E- and P-selectins, VCAM-1, ICAM-1 and PECAM-1) [Chi & Mendelez, 2007; Marchio et al, 2019; Ailuno et al., 2020], activation of the production of other pro-inflammatory cytokines, and the recruitment of immune cells to the affected arterial intima."

The suggested citations are the following:

- Chi, Z; Mendelez, AJ. Role of Cell Adhesion Molecules and Immune-Cell Migration in the Initiation, Onset and Development of Atherosclerosis. Cell Adhesion & Migration 2007, 1(4), 171-175. Doi: 10.4161/cam.1.4.5321

- Marchio, P; Guerra-Ojeda, S; Vila, JM; Aldasoro, M; Victor, VM; Mauricio, MD. Targeting Early Atherosclerosis: A Focus on Oxidative Stress and Inflammation. Oxidative Medicine and Cellular Longevity 2019, 8563845 Doi: 10.1155/2019/8563845.

- Ailuno, G; Baldassari, S; Zuccari, G; Schlich, M; Caviglioli, G. Peptide-based nanosystems for vascular cell adhesion molecule-1 targeting: a real opportunity for therapeutic and diagnostic agents in inflammation associated disorders. Journal of Drug Delivery Science and Technology 2020, 55, 101461. Doi: 10.1016/j.jddst.2019.101461.

Author Response

Point 1: Throughout the manuscript, sometimes "in vivo" and "in vitro" are in italic font, some other times don't. I suggest the authors to align the style.

Response 1: It was corrected.

Point 2: To enhance the summarising function of figure 1, I suggest the authors to add some more information next to each type of macrophages, in a schematic way. For example, which cytokines are most secreted, what triggers the activation of that particular type of macrophage, and so on.

Response 2: Factors of macrophage activation were added on figure 1.

Point 3: I suggest the authors to increase the font of figure 2, in order to ease the reading.

Response 3: The figure with increased font was added.

Point 4: On lines 272-274, when the authors state: "...which leads to an increase in the expression of various adhesion proteins, activation of the production of other pro-inflammatory cytokines, and the recruitment of immune cells to the affected arterial intima.", I suggest to mention some examples of adhesion molecules involved in these processes. Therefore, I suggest the authors to replace the sentence with: "...which leads to an increase in the expression of various adhesion proteins (like E- and P-selectins, VCAM-1, ICAM-1 and PECAM-1) [Chi & Mendelez, 2007; Marchio et al, 2019; Ailuno et al., 2020], activation of the production of other pro-inflammatory cytokines, and the recruitment of immune cells to the affected arterial intima."

The suggested citations are the following:

- Chi, Z; Mendelez, AJ. Role of Cell Adhesion Molecules and Immune-Cell Migration in the Initiation, Onset and Development of Atherosclerosis. Cell Adhesion & Migration 2007, 1(4), 171-175. Doi: 10.4161/cam.1.4.5321

- Marchio, P; Guerra-Ojeda, S; Vila, JM; Aldasoro, M; Victor, VM; Mauricio, MD. Targeting Early Atherosclerosis: A Focus on Oxidative Stress and Inflammation. Oxidative Medicine and Cellular Longevity 2019, 8563845 Doi: 10.1155/2019/8563845.

- Ailuno, G; Baldassari, S; Zuccari, G; Schlich, M; Caviglioli, G. Peptide-based nanosystems for vascular cell adhesion molecule-1 targeting: a real opportunity for therapeutic and diagnostic agents in inflammation associated disorders. Journal of Drug Delivery Science and Technology 2020, 55, 101461. Doi: 10.1016/j.jddst.2019.101461.

Response 4: It was added.

Round 2

Reviewer 2 Report

The manuscript has been well improved, and I think most of my comments have been addressed.